# Biofilm formation, *agr* typing and antibiotic resistance pattern in methicillin-resistant *Staphylococcus aureus* isolated from hospital environments

**Sabrina Sultana Rimi**[1©], **Md. Nahid Ashraf**[1©], **Sanzila Hossain Sigma**[1], **Md. Tanjir Ahammed**[1], **Mahbubul Pratik Siddique**[1], **Mohammad Ali Zinnah**[2], **Md. Tanvir Rahman**[1]*, **Md. Shafiqul Islam**[1]*

1 Department of Microbiology and Hygiene, Faculty of Veterinary Science, Bangladesh Agricultural University, Mymensingh, Bangladesh, 2 Department of Microbiology and Public Health, Faculty of Veterinary Medicine and Animal Science, Bangabandhu Sheikh Mujibur Rahman Agricultural University, Gazipur, Bangladesh

© These authors contributed equally to this work.

* shafiq_micro@bau.edu.bd (MSI); tanvirahman@bau.edu.bd (MTR)

## Abstract

Biofilm development significantly enhances the virulence of methicillin-resistant *Staphylococcus aureus* (MRSA), leading to severe infections and decreased susceptibility to antibiotics, especially in strains associated with hospital environments. This study examined the occurrence of MRSA, their ability to form biofilms, *agr* typing, and the antibiotic resistance profiles of biofilm-forming MRSA strains isolated from environmental surfaces at Mymensingh Medical College Hospital (MMCH). From 120 swab samples, 86 (71.67%) tested positive for *S. aureus*. MRSA was identified in 86 isolates using the disk diffusion technique, and by polymerase chain reaction (PCR), 56 (65.1%) isolates were confirmed to carry the *mecA* gene. The Crystal Violet Microtiter Plate (CVMP) test revealed that 80.35% (45 isolates) were biofilm-forming and 19.6% (11 isolates) were non-biofilm-forming. Out of 45 biofilm producer isolates 37.5% and 42.9% isolates exhibited strong and intermediate biofilm-forming characteristics, respectively. Molecular analysis revealed that 17.78% of MRSA isolates carried at least one gene related to biofilm formation, specifically *icaA*, *icaB*, and *icaD* genes were discovered in 13.33%, 8.89%, 6.67% of the MRSA isolates, respectively. In *agr* typing, the most prevalent group was *agr* I (71.11%), followed by group III (17.78%) and group II (11.11%). Group IV was not detected. The distribution of *agr* gene groups showed a significant difference among biofilm-forming isolates ($p < 0.05$). In *agr* group I, 18.75% of isolates carried the *icaA* gene, 12.5% carried the *icaB* gene, and 9.37% carried the *icaD* gene. Biofilm-forming genes were not detected in any of the isolates from *agr* groups II or III. There are no statistically significant differences between *agr* groups and the presence of these genes ($p > 0.05$). Antibiotic resistance varied significantly among *agr* groups, with *agr* group I displaying the highest resistance, *agr* group II, and *agr* group III exhibiting the least resistance ($p < 0.05$). Seventy-three (73.3%) of the isolates were multi-drug resistant, with *agr* group I displaying nineteen MDR patterns. The occurrence of MRSA in hospital

**Data Availability Statement:** All relevant data are within the manuscript and its Supporting information files.

**Funding:** This research was supported with funds from Ministry of Science and Technology, Government of Bangladesh (MoST. ID No. SRG 231092.). The funders had no role in study design, data collection and analysis, decision to publish, or preparation of the manuscript.

environments and their capacity to form biofilm raises concerns for public health. These findings support the importance of further research focused on *agr* quorum sensing systems as a basis for developing novel antibacterial agents.

## Introduction

The emergence of methicillin-resistant *S. aureus* (MRSA) strains is a global concern in healthcare and constitutes a significant challenge to infection control protocols [1]. MRSA causes various human illnesses, including skin infections, endocarditis, pneumonia, osteomyelitis, septic arthritis, bloodstream infections, surgical site infections, and even death, leading to significant morbidity and healthcare expenses [2]. MRSA is a Gram-positive bacterium with genetic differences from other *S. aureus* strains. Methicillin resistance in *S. aureus* is facilitated by the *mecA* gene's acquisition within its chromosomal DNA, granting resistance specifically to β-lactam antibiotics [3]. MRSA transmission primarily occurs in hospital environments through contamination of inanimate objects like bedside rails, doorknobs, and overbed tables [4].

Biofilms are described as microbial sessile communities, originating from cells firmly attached to a surface or one another [5]. Biofilm formation is a significant virulence factor in MRSA, particularly in hospital-associated strains. Biofilms contribute significantly to the pathogenesis of intravenous catheter-related bloodstream infections, contamination of medical devices, severe tissue damage, and the prolonged occurrence of nosocomial infections [6]. According to research, bacteria in biofilms may survive antibiotic doses 10 to 10,000 times greater than their free-floating counterparts [7]. Biofilm formation requires two steps: first, *S. aureus* attaches to the surface using capsular polysaccharide/adhesion (PS/A). In second step, the multilayered biofilm is formed by bacterial multiplication and is linked to polysaccharide intracellular adhesion (PIA) production. The intracellular adhesion (*ica*) locus is responsible for PIA and PS/A synthesis. The most well-characterized method for producing biofilm at the moment is PIA synthesis, which is the main component of biofilm and is encoded by the intercellular adhesion *icaADBC*-operon [8]. In addition, biofilm formation in *S. aureus* is under the control of accessory gene regulator (*agr*) quorum-sensing system [9].

Quorum sensing involves controlling gene activity based on changes in the number of cells. Quorum sensing bacteria generate and emit chemical signal molecules called autoinducers (AIs), which become more concentrated as cell density increases[10]. The *agr* operon, comprising *agrA*, *agrB*, *agrC*, and *agrD* genes, plays a pivotal role in regulating more than 70 genes within *S. aureus*. Among these, 23 genes dictate its pathogenicity and ability to cause invasive infections [11]. Besides, *S. aureus* can be classified into four distinct categories: *agr* type I to *agr* type IV [12]. The lack of activity in the *agr* system allows *Staphylococci* to initially adhere to a polystyrene surface [13]. However, it is anticipated that up regulation of *agr* system will significantly contribute to the detachment process and return cells to their planktonic state [14]. One technique for combating staphylococcal infections is to interrupt the communication mechanism (*agr*) that regulates virulence factor synthesis which can be accomplished by interfering with the signaling molecules (autoinducers) involved in *agr*, inhibiting the bacteria's ability to harm [15]. The assertion is that *agr* types exhibit distinct properties and prevalence across diverse geographical regions. Consequently, identifying prevailing types in each specific region is functionally advantageous [16]. Early diagnosis and effective control protocols against MRSA, which forms biofilms, are crucial in

combating hazardous nosocomial infections [17]. The *agr* technique is an effective molecular typing method for identifying and monitoring MRSA clones, identifying the origin of nosocomial infections in healthcare facilities, and assessing the relationship between distinct molecular types and biofilm formation [18].

There is lack of adequate data on biofilm forming MRSA from hospital environments in Bangladesh. In fact, the research on MRSA in hospital environments or hospital patients started in the previous decade, which covers mainly the prevalence of MRSA and characterization by culture, antibiotic susceptibility of MRSA, and *mecA* gene by PCR [19–21]. This study focused on assessing the occurrence of MRSA in hospital environment, their biofilm formation ability, *agr* gene typing, and revealing antibiotic resistances so that control measures could be suggested to reduce MRSA related nosocomial infection.

## Methodology

### Ethical approval

The procedures outlined in this study received approval from the Animal Welfare and Experimentation Ethics Committee at Bangladesh Agricultural University, Mymensingh [approval number AWEEC/BAU/2022(82)].

### Study region and sampling

The present research was conducted in Mymensingh Medical College & Hospital, Mymensingh Sadar, Bangladesh, from November 2022 to August 2023. Overall, 120 surface swab samples were collected aseptically, comprising medicine, surgery, and urology wards (S1 Table in S1 File). Surface samples were obtained utilizing sterile cotton buds and promptly transferred into sterile test tubes filled with 5ml of Nutrient broth (Himedia, India). After collecting and labeling, samples were transferred to the laboratory by maintaining cool chain.

### Isolation, identification, and molecular detection of *S. aureus*

Initially, the Mannitol Salt Agar (MSA) medium (HiMedia, India) was used to isolate *S. aureus*. At first, a loopful specimen of overnight-grown bacteria was streaked onto individual MSA plates. These plates were then incubated aerobically at the optimum temperature (37˚C) for 24 hours. On MSA agar plates, organisms exhibiting golden yellow colonies were suspected as *S. aureus* and sub-cultured to obtain pure colonies. Standard tests, including Gram's staining and biochemical assays (sugar fermentation, catalase, and coagulase), were conducted to identify *S. aureus* [22]. PCR was employed to confirm *S. aureus* and MRSA isolates molecularly by targeting the genes *nuc* and *mecA*, respectively, in Table 1.

### MRSA identification

Each *S. aureus* isolates were grown using disc diffusion method on Mueller-Hinton agar medium with) oxacillin (1μg/ml) and cefoxitin (30 μg/ml) as per the guideline of Clinical Laboratory Standards Institute (CLSI) [27]. The isolates were subsequently incubated for 24 hours at 37˚C. MRSA isolates were confirmed by PCR using specific primers listed in Table 1.

### Biofilm development of MRSA

**Biofilm production assessment by Crystal Violet Microtiter Plate (CVMP) assay.** The capacity of *S. aureus* to form biofilms was assessed using 96-well flat-bottomed microtiter polystyrene plates, following the methodology outlined by Kouidhi [28]. In brief, each MRSA isolate was inoculated into 5 ml of sterile tryptic soy broth (TSB), followed by an 18-hour

**Table 1. List of primers, with oligonucleotide sequences, amplicon size, annealing temperature, and references, which were used to detect various genes of *Staphylococcus aureus*.**

| List of the primers | Primer's sequence (5'-3') | Amplicon size | Annealing temperature | References |
|---|---|---|---|---|
| *nuc* F | GCGATTGATGGTGATACGGTT | 279 bp | 57 ˚C | [23] |
| *nuc* R | AGCCAAGCCTTGACGAACTAAAGC | | | |
| *mecA* F | AAAATCGATGGTAAAGGTTGGC | 533 bp | 55 ˚C | [24] |
| *mecA* R | AGTTCTGCAACTACCGGATTTTGC | | | |
| *icaA* F | GACCTCGAAGTCAATAGAGGT | 814 bp | 60 ˚C | [25] |
| *icaA* R | CCCAGTATAACGTTGGATACC | | | |
| *icaB* F | ATCGCTTAAAGCACACGACGC | 526 bp | 59 ˚C | |
| *icaB* R | TATCGGCATCTGGTGTGACAG | | | |
| *icaC* F | ATAAACTTGAATTAGTGTATT | 989 bp | 42 ˚C | |
| *icaC* R | ATATATAAAACTCTCTTAACA | | | |
| *icaD* F | AGGCAATATCCAACGGTAA | 371 bp | 59 ˚C | |
| *icaD* R | GTCACGACCTTTCTTATATT | | | |
| *pan-agr* F | ATGCACATGGTGCACATGC | | 55 ºC | [26] |
| *agr* I R | GTCACAAGTACTATAAGCTGCGAT | 440 bp | | |
| *agr* II R | GTATTACTAATTGAAAAGTGCCATAGC | 572 bp | | |
| *agr* III R | CTGTTGAAAAAGTCAACTAAAAGCTC | 406 bp | | |
| *agr* IV R | CGATAATGCCGTAATAC CCG | 588 bp | | |

incubation period at 37 ˚C without shaking. The isolates' growth was standardized using a 0.5 McFarland concentration; each strain's cell concentration was brought down to roughly $10^8$ CFU/ml [29]. The growing cultures were subsequently diluted in TSB, a culture medium, with the addition of 10% glucose using a 10-fold dilution technique. For every strain, 200 μl of the diluted culture was distributed across three wells in a microtiter plate. Subsequently, the plate underwent a twenty-four- hour incubation period at 37 ˚C. The negative control well was filled with a broth medium containing TSB and 10% glucose. The planktonic cells or bacteria were eliminated by rinsing each microtiter well using sterile PBS 3–5 times. After rinsing, the formed biofilm is immersed in 95% ethanol for 15 to 20 minutes at room temperature for fixation. Then the fixative is removed by decanting. After fixation, the biofilm is stained with 100 μl of 1% crystal violet and left for a few minutes to stain. Then, the excess stain is washed off. The optical density (OD) was measured in an automated spectrophotometer at 570 nm (VWR, part of Avantor, Radnor, PA, USA). Each isolate was classified as a strong biofilm producer ($OD_{570} \geq 1$), moderate/intermediate biofilm producer ($0.1 \leq OD_{570} < 1$), or non-biofilm producer ($OD_{570} < 0.1$) based on the results of the biofilm formation test [30].

**Genotypic analysis of biofilm formation.** Biofilm-forming MRSA strains were identified through molecular detection using PCR-based amplification of the adhesion genes within the *icaADBC* operon, namely *icaA*, *icaB*, *icaC*, and *icaD*.

## *agr* typing of biofilm-forming MRSA strains

PCR was employed to determine the *agr* types of biofilm-forming MRSA strains, utilizing the primers outlined in Table 1.

## Antibiogram study of isolated biofilm-forming MRSA strains

The disc diffusion method was utilized for detecting antimicrobial susceptibility against eight commonly used antibiotics [31], with results categorized as susceptible, intermediate, and

**Table 2. Antimicrobial agents with their disc concentration (CLSI, 2021).**

| Name of the antibiotic | Symbol | Disc concentration (μg/disc) |
|---|---|---|
| Ampicillin | AMP | 25 |
| Erythromycin | E | 15 |
| Tetracycline | TE | 10 |
| Ciprofloxacin | CIP | 5 |
| Co-Trimoxazole | COT | 25 |
| Gentamicin | GEN | 10 |
| Vancomycin | VA | 30 |
| Chloramphenicol | C | 30 |

resistant based on Clinical and Laboratory Standards Institute's (CLSI) interpretative standards [27]. Antibiotics used in this study are listed in Table 2.

## Statistical analysis

The data collected in this study were inputted into Excel 365 and then transferred to IBM SPSS 25.0 for analysis.

**Descriptive analysis.** The Wilson and Brown Hybrid method was applied to compute binomial 95% confidence intervals [32]. Additionally, differences in the occurrence of *S. aureus* and MRSA among wards, *agr* gene group distribution, prevalence of *icaA*, *icaB*, and *icaD* in different *agr* group and the antibiotic resistance patterns in *agr* types among biofilm-forming MRSA isolates were assessed using the chi-square test. Statistical significance was established for *p*-values below 0.05.

**Bivariate analysis.** Utilizing SPSS to compute Spearman's correlation coefficients with statistical significance at $p<0.05$, a bivariate study was performed to ascertain the correlation between genes linked to biofilm-forming MRSA isolates.

## Heatmap analysis

A heatmap was created to illustrate MRSA biofilm-forming genes using Excel 365, with values "1" and "0" considered positive and negative.

## Results

### Frequency of *S. aureus* and MRSA

In the analysis of 120 samples, 97 exhibited positive indications of *S. aureus* based on observing its morphological and biochemical characteristics. Using PCR, to target the *nuc* gene, 86 samples (71.67%, 95% CI: 62.88–79.12%) were confirmed positive for *S. aureus* (S1 Fig in S1 File). Notably, the surgery ward demonstrated the highest frequency of positive isolates (96%), while the urology ward exhibited the lowest frequency (65%). The final frequency of MRSA among the *S. aureus* isolates in our investigation was found to be 65.1% (56 out of 86 isolates) by both disc diffusion and the presence of the *mecA* gene (S1 Fig in S1 File). However, *S. aureus* and MRSA frequency varied between wards without statistically significant difference ($p > 0.05$), as presented in Tables 3 and 4, respectively.

### Formation of biofilm based on CVMP assay

Among the MRSA strains studied, the CVMP assay revealed that 80.35% (45 isolates) were biofilm producers, while 19.6% (11 isolates) were non-biofilm producers. Out of 45 biofilm-

Table 3. Frequency of *Staphylococcus aureus* isolated from different wards.

| Sample | Positive *S. aureus* No. (%) | 95% CI | *p*-value |
|---|---|---|---|
| Medicine ward (n = 40) | 28 (70%) | 55.80–84.20 | *p* > 0.05 |
| Surgery ward (n = 40) | 32 (80%) | 67.60–92.40 | |
| Urology ward (n = 40) | 26 (65%) | 50.22–79.78 | |
| Total (n = 120) | 86 (71.67%) | 62.88–79.12 | |

In this context, a *p*-value of 0.05 or lower ($p < 0.05$) indicated statistical significance. "n" represents the sample size, while "CI" stands for confidence interval.

Table 4. Frequency of MRSA isolated from different wards.

| *S. aureus* isolates | MRSA No. (%) | 95% CI | *p*-value |
|---|---|---|---|
| Medicine ward (n = 28) | 20 (71.4%) | 54.70–88.16 | *p* > 0.05 |
| Surgery ward (n = 32) | 19 (59.4%) | 42.36–76.39 | |
| Urology ward (n = 26) | 17 (65.4%) | 47.10–83.67 | |
| Total (n = 86) | 56 (65.1%) | 55.04–75.19 | |

In this context, a *p*-value of 0.05 or lower ($p < 0.05$) indicated statistical significance. "n" represents the sample size, while "CI" stands for confidence interval.

producing MRSA isolates, 37.5% (95% CI: 24.82–50.18%) exhibited strong biofilm-forming characteristics, while 42.9% (95% CI: 29.90–55.82%) showed intermediate characteristics (Fig 1). Notably, the surgery ward displayed the highest occurrence of strong biofilm-forming MRSA isolates at 52.63%, according to the CVMP test results.

## Formation of biofilm based on genotypic characteristics

Through PCR analysis, it was determined that among 45 isolates exhibiting phenotypic biofilm positivity, 8 (17.78%, 95% CI: 7.76–27.79%) harbored at least one gene related to biofilm formation. Notably, the occurrence of the *icaA* gene was highest at 13.33% (95% CI: 4.43–22.24%), followed by the *icaB* gene at 8.89% (95% CI: 1.44–16.34%) and the *icaD* gene at 6.67%

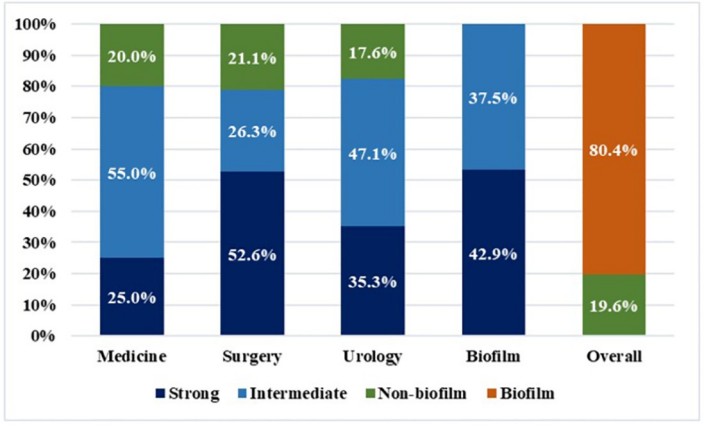

Fig 1. Occurrence of biofilm-forming MRSA isolated from different wards.

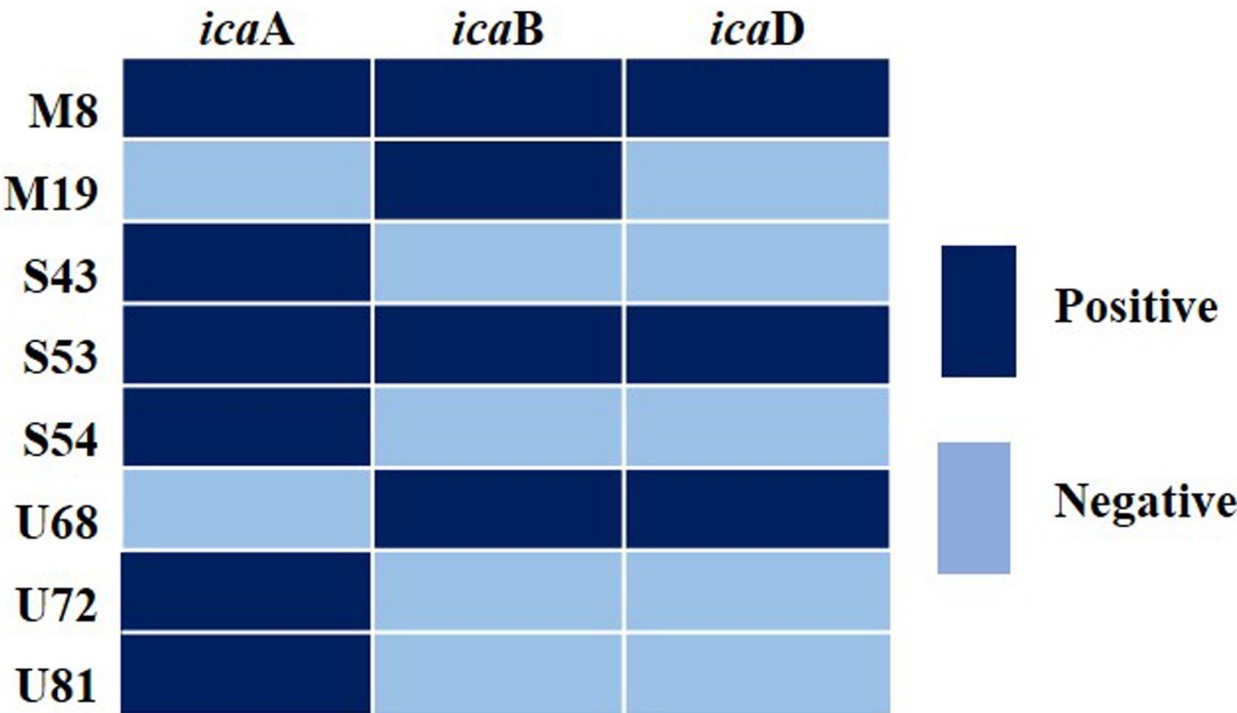

**Fig 2. Heatmap depicting the presence of genes associated with biofilm formation in MRSA strains collected from various wards of MMCH, M = Medicine, S = Surgery, U = Urology.**

(95% CI: 0.13–13.20%) (S1 Fig in S1 File). The *icaC* gene was not detected in any of the isolates. Out of 8 genotypically biofilm-positive isolates, two isolates carried three adhesion genes, while one isolate possessed two genes (Fig 2).

In the bivariate analysis, strong positive correlations were identified among the biofilm-forming genes, with correlation coefficients (ρ values) of 0.567 ($p < 0.001$) for the pair *icaA* and *icaB*, 0.419 ($p = 0.004$) for *icaA* and *icaD*, and 0.543 ($p < 0.001$) for *icaB* and *icaD* (Table 5).

### Biofilm-forming MRSA isolates and *agr* locus distribution

Concerning the distribution of *agr* loci, among the 45 biofilm-forming MRSA isolates, 32 strains (71.11%) were classified under *agr* group I, while 5 strains (11.11%) were classified

**Table 5. Spearman's correlation coefficients (ρ) calculated for adhesion gene pairs to evaluate biofilm-forming MRSA isolates from different wards of MMCH (n = 45).**

|  |  | *icaA* | *icaB* | *icaD* |
|---|---|---|---|---|
| *icaA* | Spearman's Correlation | 1 |  |  |
|  | Sig. (2-tailed) | - |  |  |
| *icaB* | Spearman's Correlation | .567** | 1 |  |
|  | Sig. (2-tailed) | .001 | - |  |
| *icaD* | Spearman's Correlation | .419** | .543** | 1 |
|  | Sig. (2-tailed) | .004 | .001 | - |

** The correlation is statistically significant at the 0.01 significance level. (2-tailed).

**Table 6. Biofilm-forming MRSA isolates and distribution of *agr* loci.**

| Biofilm production (N)% | *agr* gene | Isolates(N)% | *p*-value |
|---|---|---|---|
| **(45/56) 80.35** | I | (32/45) 71.11 | $p < 0.05$ |
| | II | (5/45) 11.11 | |
| | III | (8/45) 17.78 | |
| | IV | 0 | |

In this context, a *p*-value of 0.05 or lower ($p < 0.05$) indicated statistical significance. "N" represents the sample size.

under *agr* group II, and 8 strains (17.78%) were categorized under *agr* group III through PCR analysis (S1 Fig in S1 File). Notably, *agr* group IV was not identified in any of the examined biofilm-forming strains. A statistically significant difference was observed in the distribution of *agr* gene types among biofilm-forming isolates ($p < 0.05$) (Table 6). Within *agr* group I, 18.75% of isolates possessed the *icaA* gene, 12.5% possessed the *icaB* gene, and 9.37% possessed the *icaD* gene. None of the isolates possessing biofilm-forming genes were classified under *agr* group II or group III. There is no statistically significant differences between *agr* groups and the presence of these genes ($p > 0.05$) (Table 7).

## Antibiotic resistance profile of MRSA (biofilm-forming) isolates

Various antimicrobial agents were selected to assess the susceptibility patterns of 45 biofilm-forming isolates within specific *agr* groups. The resistance percentages for the tested antimicrobial agents were determined for each *agr*-specific group. In our investigation, it was observed that all strains exhibited high resistance (100%) to both oxacillin and cefoxitin while demonstrating complete sensitivity (100%) to vancomycin. The susceptibility of the studied isolates to other antimicrobial agents varied, with approximately 62.22% of strains resistant to ampicillin, 42.22% to erythromycin, 31.11% to gentamicin, 24.44% to chloramphenicol, 8.88% to tetracycline, 51.11% to co-trimoxazole, and 40.00% to ciprofloxacin. The highest prevalence of antibiotic resistance was notably observed in *agr* group I, followed by *agr* group III. In contrast, *agr* group II exhibited the least resistance, showing a statistically significant difference ($p < 0.05$) (Table 8).

A significant percentage of MRSA strains showed multidrug resistance, with 73.3% of the isolates displaying resistance to three or more antibiotics. The multidrug-resistant strains exhibited a variety of resistance levels, ranging from resistance to 3 classes of antibiotics (8 strains, 17.77%) to resistance to 8 classes of antibiotics (1 strain, 2.2%). Nine strains (20%) among the studied strains exhibited the most prevalent resistance pattern, showing resistance against 4 to 5 classes of antibiotics. Table 9 contains detailed resistance patterns for various antibiotics.

**Table 7. Prevalence of *icaA*, *icaB* and *icaD* in different *agr* group.**

| *agr* group | No. of isolates (N) | *icaA* (%) | *icaB* (%) | *icaD* (%) | *p*-value |
|---|---|---|---|---|---|
| *agr* I | 32 | 6 (18.75) | 4 (12.5) | 3 (9.37) | $p > 0.05$ |
| *agr* II | 5 | 0 | 0 | 0 | |
| *agr* III | 8 | 0 | 0 | 0 | |

In this context, a *p*-value of 0.05 or lower ($p < 0.05$) indicated statistical significance. "N" represents the sample size.

**Table 8. Antibiotic resistance patterns in MRSA biofilm-forming isolates across *agr* specific groups (N = 45).**

| Antibiotics | *agr* I (N = 32) | *agr* II (N = 5) | *agr* III (N = 8) | % resistance (N = 45) | *p*-value |
|---|---|---|---|---|---|
| Ampicillin | 28(87.5%) | 0 | 0 | 28(62.22%) | *p* < 0.05 |
| Erythromycin | 17(53.12%) | 1(20%) | 1(12.5%) | 19(42.22%) | |
| Gentamicin | 14(43.75%) | 0 | 0 | 14(31.11%) | |
| Methicillin | 32(100%) | 5(100%) | 8(100%) | 45(100%) | |
| Vancomycin | 0 | 0 | 0 | 0 | |
| Cefoxitin | 32(100%) | 5(100%) | 8(100%) | 45(100%) | |
| Chloramphenicol | 10(31.25%) | 0 | 1(12.5%) | 11(24.44%) | |
| Tetracycline | 4(12.5%) | 0 | 0 | 4(8.88%) | |
| Co-trimoxazole | 23(71.87%) | 0 | 0 | 23(51.11%) | |
| Ciprofloxacin | 18(56.25%) | 0 | 0 | 18(40.00%) | |

In this context, a *p*-value of 0.05 or lower ($p < 0.05$) indicated statistical significance. "N" represents the sample size.

**Table 9. Multi-drug resistance (MDR) pattern of MRSA biofilm-forming isolates (N = 45).**

| No. of patterns | Antibiotic resistance patterns | No. of MDR isolates | (%) | *agr* types |
|---|---|---|---|---|
| 1 | AMP,E,GEN,FOX,C,TE,COT,CIP | 1 | 2.2 | I |
| 2 | AMP,E,GEN,FOX,C,COT | 1 | 13.33 | I |
| 3 | AMP,E,GEN,FOX,COT,CIP | 3 | | I |
| 4 | AMP,GEN,FOX,C,TE,COT | 1 | | I |
| 5 | AMP,GEN,FOX,C,TE,CIP | 1 | | I |
| 6 | AMP,E,FOX,C,COT | 2 | 20 | I |
| 7 | AMP,GEN,FOX,C,COT | 2 | | I |
| 8 | AMP,FOX,C,TE,CIP | 1 | | I |
| 9 | AMP,E,GEN,FOX,COT | 1 | | I |
| 10 | AMP,FOX,C,COT,CIP | 1 | | I |
| 11 | AMP,E,FOX,COT,CIP | 2 | | I |
| 12 | AMP,FOX,COT,CIP | 4 | 20 | I |
| 13 | GEN,FOX,COT,CIP | 1 | | I |
| 14 | AMP,E,FOX,CIP | 3 | | I |
| 15 | AMP,GEN,FOX,COT | 1 | | I |
| 16 | E,FOX,CIP | 1 | 17.77 | I |
| 17 | AMP,E,FOX | 3 | | I |
| 18 | E,FOX,C | 1 | | III |
| 19 | GEN,FOX,COT | 2 | | I |
| 20 | AMP,FOX,COT | 1 | | I |
| **Total** | | 33(73.3) | | |

AMP: Ampicillin, E: Erythromycin, TE: Tetracycline, CIP: Ciprofloxacin, COT: Co-Trimoxazole, GEN: Gentamicin, C: Chloramphenicol, FOX: Cefoxitin

## Discussion

The study found a high occurrence of *S. aureus* (71.67%) and MRSA (65.1%) in the hospital environment. Comparisons to prior studies in Bangladesh revealed varying rates [19, 20, 33], emphasizing continuous monitoring. Overseas investigations demonstrated diverse MRSA detection rates [34, 35]. Differences in MRSA prevalence was attributed to location, timeframe,

infection control measures, and treatments. The findings underscore the significance of ongoing monitoring and infection control in healthcare.

The study investigated MRSA biofilm formation ability through phenotypic and genotypic analyses, utilizing CVMP method and gene amplification (*icaA*, *icaB*, *icaC*, and *icaD*). Results showed variation in biofilm-forming characteristics, with 80.35% biofilm producers (37.5% strong and 42.9% intermediate) and 19.6% non-biofilm producers among the MRSA isolates. Genotypically, 17.78% of MRSA isolates possessed not less than one biofilm-related gene, with *icaA* being the most prevalent at 13.33%, followed by *icaB* (8.89%) and *icaD* (6.67%). In our study, we did not find any MRSA isolates containing *icaC*. A comparison with studies from Bangladesh [30] and abroad [36] revealed differing prevalence rates and gene distribution, highlighting the need for region-specific insights into MRSA biofilm formation. The present study showed that the occurrence rates of genes associated with biofilm formation were notably lower than those observed in CVMP tests. Previous research noted similar discoveries [37]. The significant variability in biofilm formation indicated in our study suggests that isolates can generate biofilm without relying on *ica* gene. These findings emphasize the significance of phenotypic and genotypic assessments in various geographic scenarios and further our understanding of MRSA biofilm-forming features. The disparities observed between the current study and prior research may be ascribed to variations in sample origins, sizes, types, and geographic locations.

In our study *agr* typing of MRSA isolates revealed that the majority belonged to *agr* group I (71.11%), followed by *agr* group III (17.78%), *agr* group II (11.11%) with no detection of *agr* group IV in our strains. This distribution suggests that the prevalent *agr* type among MRSA strains is type I. Our findings align with other research, where *agr* group I was identified as the predominant type in MRSA strains[16, 38]. Notably, a higher prevalence of *agr* group II was observed in nosocomial infections [39]. Consistent with several previous studies [12, 39], we did not detect *agr* group IV in our research. These variations in *agr* group distribution may be attributed to ecological and geographical factors and differences in infection control procedures. In our study, we found that all MRSA isolates with biofilm-forming genes belonged to *agr* group I. Researchers have demonstrated that the *agr* system significantly affects the genes involved in biofilm formation. When the bacterial population reaches a specific threshold, the *agr* system is triggered, resulting in the production of RNAIII, a regulatory RNA molecule. RNAIII subsequently decreases the expression of the *ica* operon, which reduces the production of PIA and thereby inhibits biofilm formation [40, 41].

In this study the possible correlation between specific *agr* groups and antibiotic resistance in MRSA isolates were also investigated. Group I isolates displayed varying resistance to all antibiotics tested, while groups II and III were mostly susceptible, except for specific antibiotics (oxacillin and cefoxitin). The results suggest a method to enhance the sensitivity of biofilm-forming MRSA by activating the *agr* quorum-sensing system, particularly *agr* II and *agr* III. This activation, potentially using autoinducing peptides (AIP), could inhibit biofilm formation, limit biofilm growth, reduce biofilm size, or promote the detachment of MRSA from established biofilms. The findings align with a study by Boles [42], indicating that *agr* system activation could disrupt biofilm formation, making MRSA more susceptible to antibiotics. This can be attributed to the reduction in the expression of surface adhesins via activation of the *agr* system in established biofilms, which initiates a dispersal pathway. This pathway detaches cells from the surface-bound biofilm, reverting them to a planktonic state that is susceptible to antibiotics. Group I isolates also displayed 19 multidrug resistance patterns, ranging from resistance to 3–8 classes of antibiotics. The connection between MDR patterns, biofilm-forming genes, and the *agr* system is intricate and interrelated. The *agr* system is essential in regulating biofilm formation, which in turn influences the MDR phenotype. The *agr* system

has a crucial function in regulating biofilm development and, therefore, the multi-drug resistant (MDR) phenotype. [43]. The study underscores the potential of exploring quorum sensing control in MRSA to develop innovative antibacterial agents. Therefore, the major limitation of the study is the use of less number of samples, in addition, all the samples were from one location. Moreover, no sequencing of *mecA* gene was done for any MSRA isolates.

## Conclusion

In conclusion, the study's findings showed a high prevalence of *S. aureus* in hospital environments, with a notable proportion of MRSA. A significant proportion of the isolates exhibited both strong and intermediate biofilm production capabilities. The most prevalent *agr* group identified was I, followed by groups III and II. The *agr* groups differed significantly in terms of antibiotic resistance, with group I showing the highest levels of resistance and MDR pattern. We found a positive correlation between antibiotic resistance and biofilm development. Detection of biofilm forming MRSA in hospital environment is very alarming from public health point of view, because of the possibilities for transmission of these MSRA from hospital environments to patients and other visitor. Further detailed studies are required to understand the transmission dynamic of these MRSA in hospital for control measures and on the *agr* quorum sensing system for developing novel antibacterial agents. It will be good to do whole genome analysis of selected MRSA isolates for detailed resistome analysis, along with whole genome based phylogenetic analysis, to know the relatedness of the isolates to other MRSA isolates.

## Supporting information

**S1 File.**
(RAR)

## Acknowledgments

The authors would like to thank the authority who helped in sampling procedure.

## Author Contributions

**Conceptualization:** Md. Shafiqul Islam.

**Data curation:** Sabrina Sultana Rimi.

**Formal analysis:** Sabrina Sultana Rimi, Md. Nahid Ashraf, Md. Tanvir Rahman.

**Investigation:** Sabrina Sultana Rimi, Sanzila Hossain Sigma, Md. Tanjir Ahammed, Md. Tanvir Rahman, Md. Shafiqul Islam.

**Methodology:** Sabrina Sultana Rimi, Md. Nahid Ashraf, Sanzila Hossain Sigma, Md. Tanjir Ahammed, Md. Shafiqul Islam.

**Project administration:** Md. Shafiqul Islam.

**Resources:** Md. Shafiqul Islam.

**Supervision:** Mohammad Ali Zinnah, Md. Tanvir Rahman, Md. Shafiqul Islam.

**Validation:** Mahbubul Pratik Siddique, Mohammad Ali Zinnah, Md. Shafiqul Islam.

**Visualization:** Mahbubul Pratik Siddique, Mohammad Ali Zinnah, Md. Shafiqul Islam.

**Writing – original draft:** Sabrina Sultana Rimi, Mahbubul Pratik Siddique, Md. Tanvir Rahman.

**Writing – review & editing:** Mahbubul Pratik Siddique, Mohammad Ali Zinnah, Md. Tanvir Rahman, Md. Shafiqul Islam.

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
