## [Decision Letter · Decision Letter 0]

15 May 2024

PONE-D-24-11847Biofilm formation, agr typing and antibiotic resistance pattern in methicillin-resistant Staphylococcus aureus isolated from hospital environmentsPLOS ONE

Dear Dr. Islam,

Thank you for submitting your manuscript to PLOS ONE. After careful consideration, we feel that it has merit but does not fully meet PLOS ONE’s publication criteria as it currently stands. Therefore, we invite you to submit a revised version of the manuscript that addresses the points raised during the review process.

We look forward to receiving your revised manuscript.

Kind regards,

Seyed Mostafa Hosseini

Academic Editor

PLOS ONE

Journal Requirements:

3. We note that [Figure 1] in your submission contain [map/satellite] images which may be copyrighted. All PLOS content is published under the Creative Commons Attribution License (CC BY 4.0), which means that the manuscript, images, and Supporting Information files will be freely available online, and any third party is permitted to access, download, copy, distribute, and use these materials in any way, even commercially, with proper attribution. For these reasons, we cannot publish previously copyrighted maps or satellite images created using proprietary data, such as Google software (Google Maps, Street View, and Earth). For more information, see our copyright guidelines: http://journals.plos.org/plosone/s/licenses-and-copyright.

Reviewers' comments:

Reviewer's Responses to Questions

**Comments to the Author**

1. Is the manuscript technically sound, and do the data support the conclusions?

Reviewer #1: Yes

Reviewer #2: Yes

2. Has the statistical analysis been performed appropriately and rigorously? 

Reviewer #1: No

Reviewer #2: Yes

3. Have the authors made all data underlying the findings in their manuscript fully available?

Reviewer #1: No

Reviewer #2: Yes

4. Is the manuscript presented in an intelligible fashion and written in standard English?

Reviewer #1: Yes

Reviewer #2: Yes

5. Review Comments to the Author

Reviewer #1: Dear Editorial Board of the PLOS ONE journal,

I have reviewed the manuscript entitled " Biofilm formation, agr typing and antibiotic resistance pattern in methicillin-resistant Staphylococcus aureus isolated from hospital environments " and would like to provide my feedback as follows:

Comment 1: Please delete the point at the end of the manuscript title.

Comment 2: In line 17, "MRSA's occurrence" should be replace with a better phrase, and the sentence should be rewritten.

Comment 3: In line 21, "PCR" should be written in full form for first time.

Comment 4: In line 21 and throughout the manuscript "mecA" should be written in italic form.

Comment 5: In line 25 and throughout the manuscript " icaA, icaB, and icaD" should be written in italic form.

Comment 6: In line 30, Please follow the order of the group II and III.

Comment 7: In line 22 and 23, please classified the isolates into biofilm forming and non-biofilm forming. Then the biofilm forming isolates classified into strong, moderate, and weak patterns. Please separate these two categories from each other's throughout the manuscript (following diagrams).

Comment 8: In line 39, please delete "Staphylococcus aureus" and only use "MRSA". The full name is already mentioned. Please check this point throughout the text.

Comment 9: In line 42, please replace "gram-positive" with "Gram-positive". Gram is the name of a scientist and must start with a capital letter.

Comment 10: In line 47, please add the comma after "communities".

Comment 11: In line 58, please delete "polysaccharide intercellular adhesin" and use "PIA".

Comment 12: In line 60, please replace "agr (accessory gene regulator)" with "accessory gene regulator (agr)". Please check all abbreviations and full names throughout the manuscript.

Comment 13: In line 63 and 64, please rewritten the sentence.

Comment 14: In line 71, please replace " Staphylococci" to "staphylococcal", in not italic form.

Comment 15: In line 86, please add the comma after " typing".

Comment 16: In line 103, please use "MSA" after "Mannitol Salt Agar".

Comment 17: In line 110, please specify genes required for approving S. aureus and MRSA isolates, separately.

Comment 18: In line 116, please replace " methicillin" with " oxacillin (1 μg) ".

Comment 19: In line 116, according to CLSI 2021, the approved method to detection of MRSA is: 1)MIC for oxacillin MIC, 2) MIC and disc diffusion for cefoxitin. Please justify this point for MRSA detection.

Comment 20: In line 124, please add "TSB" after "tryptic soy broth".

Comment 21: In line 132, after washing the formed biofilms, please explain what method was used to fixation them.

Comment 22: In line 135, please mention the manufacture company and other detail about of the device (ELISA reader).

Comment 23: In line 136, Please use the following formula to classify biofilms and change your data accordingly: OD < ODc (non-biofilm producer), ODc < OD < 2xODc (weak biofilm producer), 2xODc < OD < 4xODc (moderate biofilm producer), 4xODc < OD (strong biofilm producer).

Comment 24: In line 148: Please use the full name of "CLSI".

Comment 25: In section "Antibiogram Study of Isolated Biofilm-forming MRSA Strains", streptomycin is not recommended against S. aureus isolates according to CLSI guidelines. Please justify that why this antibiotic was chosen for antibiogram susceptibility test?

Comment 26: In section " Descriptive Analysis", please justify that why confidence intervals were considered 95%?

Comment 27: In section "Bivariate Analysis", please define other statistical tests used in the study, and more complete this section.

Comment 29: In line 172, please replace " Staphylococcus aureus" with "S. aureus".

Comment 30: In table 3 and 4, please define the "medicine ward". The urology and Surgery are not medicine ward??? Please also edit this point in figure 3, and throughout the manuscript.

Comment 31: In line 176 and 177, please define the final frequency of MRSA among S. aureus in this study. MRSA is confirmed by both disc diffusion and the presence of mecA gene.

Comment 32: In table 3 and 4, please justify that why P- value was calculated?

Comment 33: In line 187-190, please categorize the isolates firstly into biofilm forming and non-biofilm forming isolates. Then, the biofilm forming isolates should be classified into classes: strong, moderate, and weak. Please add a table about these categories (according to comment 7).

Comment 34: In tables 6 and 7, please add in a discription about statistical significance and confidence intervals (as described in table 3,4).

Comment 35: Please add a table about the prevalence icaA, B, C, D in different agr groups, which was one the main goals of this study. Please calculate the P-value about these data (as presented in table 6).

Comment 36: please justify that why authors did not determine the prevalence of MDR pattern among MRSA isolates. The investigation of relationship between MDR pattern and biofilm forming genes and agr system could more increase the scientific value of study.

Comment 37: In discussion, please deeply discus about the impact of agr system on biofilm-forming genes (icaA,B,C,D) and antibiotic resistance.

Comment 38: In my opinion, one of the main deficiencies in this article is the lack of attention to the expression levels of biofilm forming genes, which, if included, can have a significant impact on the scientific quality of the paper. Because presence or absence of these genes can influence S. aureus antibiotic resistance and biofilm formation, nonetheless, but the expression levels of these genes can vary due to environmental conditions and regulatory factors. Therefore, even if the genes are present, their expression levels can fluctuate, leading to differences in QS-mediated behaviors, such as biofilm formation and virulence factor production. To determine their expression levels in clinical strains of S. aureus, molecular techniques like quantitative gene expression analysis (qPCR) can be employed.

Comment 39: While the discussion highlights discrepancies in antibiotic resistance patterns and agr system, it does not thoroughly discuss the limitations of the study. It's essential to acknowledge potential limitations, such as sample size, selection bias, or methodological constraints, to provide a balanced interpretation of the results.

Comment 40: Consider adding a brief section in the discussion that outlines potential future research directions or practical implications of the study's findings. This can help readers understand the broader significance of the research.

Reviewer #2: 1. figure 1 is not necessary, it is better to remove it (a graph of the frequency of antibiotic resistance can be added instead).

2. What is meant by medicine section in table number 3

3. In the conclusion section, the results should not be explained again, but a general conclusion of the findings of this study should be presented and explanations regarding the relationship between agr and biofilm and antibiotic resistance should be added in this section.

6. PLOS authors have the option to publish the peer review history of their article (what does this mean?). If published, this will include your full peer review and any attached files.

Reviewer #1: No

Reviewer #2: No

---

## [Author Response · Author response to Decision Letter 0]

14 Jul 2024

Response to the comment of Reviewer #1:

Comment 1: Please delete the point at the end of the manuscript title.

Reply: Thank you for your observation. We have removed the point at the end of the manuscript title as per your suggestion.

Comment 2: In line 17, "MRSA's occurrence" should be replace with a better phrase, and the sentence should be rewritten.

Reply: Thank you for your feedback. We agree that "MRSA's occurrence" could be improved for clarity. We have revised the sentence in line 17 to enhance readability and precision.

Comment 3: In line 21, "PCR" should be written in full form for first time.

Reply: Thank you for your suggestion. We have revised line 21 to write out "Polymerase Chain Reaction (PCR)" in full form for the first mention. Subsequent mentions will use the abbreviation "PCR."

Comment 4: In line 21 and throughout the manuscript "mecA" should be written in italic form.

Reply: Thank you for highlighting the formatting issue. We have revised line 21 and ensured that "mecA" is written in italic form throughout the manuscript.

Comment 5: In line 25 and throughout the manuscript " icaA, icaB, and icaD" should be written in italic form.

Reply: Thank you for your attention to the formatting of gene names. We have revised line 25 and ensured that "icaA, icaB, and icaD" are written in italic form throughout the manuscript.

Comment 6: In line 30, Please follow the order of the group II and III.

Reply: Thank you for pointing out this. We have revised line 30 to follow the correct order of groups II and III as per your suggestion.

Comment 7: In line 22 and 23, please classified the isolates into biofilm forming and non-biofilm forming. Then the biofilm forming isolates classified into strong, moderate, and weak patterns. Please separate these two categories from each other's throughout the manuscript (following diagrams). 

Reply: Thank you for your insightful suggestion. We have revised lines 22 and 23 to classify the isolates into biofilm forming and non-biofilm forming categories. Furthermore, the biofilm forming isolates are now categorized into strong, moderate, and weak patterns. We have ensured that these two categories are separated throughout the manuscript.

Comment 8: In line 39, please delete "Staphylococcus aureus" and only use "MRSA". The full name is already mentioned. Please check this point throughout the text. 

Reply: Thank you for your valuable feedback. We have deleted "Staphylococcus aureus" and used only "MRSA" in line 39 as the full name was already mentioned previously. We have also ensured consistency throughout the text regarding this point.

Comment 9: In line 42, please replace "gram-positive" with "Gram-positive". Gram is the name of a scientist and must start with a capital letter.

Reply: Thank you for your attention to detail. We appreciate your thorough review. The correction has been made as per your suggestion, and "gram-positive" has been changed to "Gram-positive" in line 42 to appropriately capitalize the name of the scientist.

Comment 10: In line 47, please add the comma after "communities".

Reply: Thank you for your attention to detail. We have added the comma after "communities" in line 47 as per your suggestion.

Comment 11: In line 58, please delete "polysaccharide intercellular adhesin" and use "PIA".

Reply: Thank you for your comment. We have deleted "polysaccharide intercellular adhesin" and used "PIA" instead in line 58, as per your suggestion.

Comment 12: In line 60, please replace "agr (accessory gene regulator)" with "accessory gene regulator (agr)". Please check all abbreviations and full names throughout the manuscript. 

Reply: Thank you for your suggestion. We have replaced "agr (accessory gene regulator)" with "accessory gene regulator (agr)" in line 60, and we have ensured consistency in the use of abbreviations and full names throughout the manuscript.

Comment 13: In line 63 and 64, please rewritten the sentence.

Reply: Thank you for your feedback. We have revised lines 63 and 64 to improve clarity and readability of the sentence.

Comment 14: In line 71, please replace " Staphylococci" to "staphylococcal", in not italic form.

Reply: Thank you for your comment. We have replaced "Staphylococci" with "staphylococcal" (not italicized) in line 71, as per your suggestion.

Comment 15: In line 86, please add the comma after " typing".

Reply: Thank you for your suggestion. We have added the comma after "typing" in line 86 for improved clarity.

Comment 16: In line 103, please use "MSA" after "Mannitol Salt Agar".

Reply: Thank you for your suggestion. We have revised line 103 to use "MSA" after "Mannitol Salt Agar", as per your recommendation.

Comment 17: In line 110, please specify genes required for approving S. aureus and MRSA isolates, separately.

Reply: Thank you for your feedback. We have specified the genes required for approving S. aureus and MRSA isolates separately in line 110, as per your suggestion.

Comment 18: In line 116, please replace " methicillin" with " oxacillin (1 μg) ".

Reply: Thank you for your comment. We have replaced "methicillin" with "oxacillin (1 μg)" in line 116, as per your suggestion.

Comment 19: In line 116, according to CLSI 2021, the approved method to detection of MRSA is: 1)MIC for oxacillin MIC, 2) MIC and disc diffusion for cefoxitin. Please justify this point for MRSA detection.

Reply: Thank you for your valuable feedback on our manuscript. We appreciate the opportunity to clarify our methodology regarding the detection of MRSA. In our study, we have adhered to the CLSI 2021 guidelines to the best of our capabilities. Specifically, the guidelines recommend the use of oxacillin MIC, as well as both MIC and disc diffusion methods for cefoxitin, for the detection of MRSA. However, due to the lack of facilities for performing MIC tests in our laboratory, we employed the disc diffusion method for both oxacillin and cefoxitin. While the MIC method is preferred for its precision, the disc diffusion method remains a widely accepted and standardized technique for the detection of antibiotic resistance, including MRSA. The CLSI provides well-defined interpretative criteria for both oxacillin and cefoxitin disc diffusion tests. The disc diffusion method is simpler, cost-effective, and does not require specialized equipment, making it feasible for our laboratory setting. This method allows us to efficiently screen a large number of isolates. To ensure the accuracy of our MRSA detection, we supplemented the phenotypic disc diffusion tests with molecular detection of the mecA gene. The presence of the mecA gene is a definitive marker for MRSA. Although our laboratory lacks the facilities to perform MIC tests, the use of disc diffusion methods for both oxacillin and cefoxitin, combined with the molecular detection of the mecA gene, allows us to effectively and accurately detect MRSA. This approach is consistent with CLSI guidelines to the extent possible and provides a reliable alternative for MRSA detection in resource-limited settings. We hope this explanation addresses your concerns and justifies our methodology. We are committed to adhering to the highest standards in our research and appreciate your understanding of the constraints we face.

Comment 20: In line 124, please add "TSB" after "tryptic soy broth".

Reply: Thank you for your thorough review and valuable feedback on our manuscript. Regarding your suggestion to add an abbreviation for "tryptic soy broth" in line 124, we have updated the manuscript accordingly.

Comment 21: In line 132, after washing the formed biofilms, please explain what method was used to fixation them. 

Reply: Thank you for your inquiry regarding the fixation method used in our study, particularly in line 132. Following the washing step, we employed a fixation procedure involving immersion of the formed biofilms in 95% ethanol for 15 to 20 minutes at room temperature. This method was chosen to effectively stabilize the biofilm structure for subsequent analysis.

Comment 22: In line 135, please mention the manufacture company and other detail about of the device (ELISA reader).

Reply: Thank you for your query regarding the ELISA reader mentioned in line 135 of our manuscript. The device used for our study was the VWR ELISA reader, provided by VWR, part of Avantor, Radnor, PA, USA. We appreciate the opportunity to provide this additional information.

Comment 23: In line 136, Please use the following formula to classify biofilms and change your data accordingly: OD < ODc (non-biofilm producer), ODc < OD < 2xODc (weak biofilm producer), 2xODc < OD < 4xODc (moderate biofilm producer), 4xODc < OD (strong biofilm producer).

Reply: Thank you for your valuable feedback and for suggesting a specific formula for classifying biofilms. In our study, we followed the classification methodology outlined by Ballah et al. (2022), where biofilm producers were categorized as follows: Strong biofilm producer: OD570 ≥ 1, Moderate/intermediate biofilm producer: 0.1 ≤ OD570 < 1, Non-biofilm producer: OD570 < 0.1. We chose to adhere to this classification system to maintain consistency with the established protocol in Ballah et al.'s work, which is widely recognized and accepted within our research community. This consistency is crucial for ensuring that our findings are directly comparable with other studies that have utilized the same methodology, thereby enhancing the reliability and relevance of our results. We acknowledge that the classification formula you provided—OD < ODc (non-biofilm producer), ODc < OD < 2xODc (weak biofilm producer), 2xODc < OD < 4xODc (moderate biofilm producer), and 4xODc < OD (strong biofilm producer)—is a well-recognized approach. However, our classification based on OD570 values provides a clear and effective way to categorize biofilm production levels as per Ballah et al. (2022). Given the precedence and acceptance of Ballah et al.'s criteria in related research, we believe our classification method is justified. Thank you once again for your insightful comments and for helping us improve our work.

Comment 24: In line 148: Please use the full name of "CLSI".

Reply: Thank you for your meticulous review and for pointing out the need to use the full name of "CLSI" in line 148 of our manuscript. We have revised the text to include the full name, "Clinical and Laboratory Standards Institute (CLSI) ". We appreciate your attention to detail and your efforts to improve the quality of our manuscript. 

Comment 25: In section "Antibiogram Study of Isolated Biofilm-forming MRSA Strains", streptomycin is not recommended against S. aureus isolates according to CLSI guidelines. Please justify that why this antibiotic was chosen for antibiogram susceptibility test?

Reply: Thank you for your insightful feedback on our manuscript and for highlighting the issue regarding the inclusion of streptomycin in the antibiogram susceptibility test for S. aureus isolates. We appreciate your attention to the guidelines set by the Clinical and Laboratory Standards Institute (CLSI), which indeed do not recommend the use of streptomycin for S. aureus isolates. Upon reviewing our manuscript, we realized that the mention of streptomycin in Table 2 was an error. We apologize for this oversight and confirm that streptomycin was not actually used in our research. We have corrected Table 2 to accurately reflect the antibiotics tested in our study, excluding streptomycin. We sincerely appreciate your diligence in pointing out this discrepancy, which has allowed us to improve the accuracy and clarity of our manuscript.

Comment 26: In section " Descriptive Analysis", please justify that why confidence intervals were considered 95%?

Reply: Thank you for your insightful feedback regarding the use of 95% confidence intervals in the "Descriptive Analysis" section of our manuscript. The 95% confidence level is a conventionally accepted threshold in most scientific disciplines. The 95% confidence level offers an optimal balance between the certainty of the interval and its width. It implies that there is a 95% probability that the true parameter lies within the confidence interval, providing a reasonable level of certainty without excessively widening the interval, which would reduce precision. Utilizing a 95% confidence interval aligns our study with the majority of existing literature, enhancing the comparability of our results with those of other studies in the field. We hope this explanation clarifies the rationale behind our choice of using 95% confidence intervals in our descriptive analysis.

Comment 27: In section "Bivariate Analysis", please define other statistical tests used in the study, and more complete this section.

Reply: Thank you for your insightful feedback regarding the "Bivariate Analysis" section of our manuscript. In our study, the primary statistical test employed for bivariate analysis was the computation of Spearman's correlation coefficients. We did not perform any additional bivariate statistical tests beyond Spearman's correlation in our study.

Comment 29: In line 172, please replace " Staphylococcus aureus" with "S. aureus".

Reply: Thank you for your attentive review. We have made the correction as suggested, replacing "Staphylococcus aureus" with "S. aureus" in line 172.

Comment 30: In table 3 and 4, please define the "medicine ward". The urology and Surgery are not medicine ward??? Please also edit this point in figure 3, and throughout the manuscript.

Reply: Thank you for your insightful feedback on our manuscript and for seeking clarification regarding the definition of "medicine ward" in Tables 3 and 4. In the context of our study conducted in Bangladesh, we use specific definitions for different hospital wards based on the type of care provided. In Bangladesh, the term "medicine ward" specifically refers to a hospital ward where patients are treated for conditions that can be managed and cured solely through medical (non-surgical) treatments. These patients typically have internal medical conditions such as infections, chronic diseases, or other non-surgical illnesses. The "surgery ward" is designated for patients who have undergone surgical procedures. These patients require post-operative care and recovery services provided by the surgical team. The ward focuses on managing the post-surgical recovery process, including wound care, pain management, and monitoring for surgical complications. The "urology ward" caters specifically to patients with urological conditions or those who have undergone surgeries related to the urinary system. This ward provides specialized care for conditions such as kidney stones, urinary tract infections, prostate issues, and post-operative care for urological surgeries. We hope this explanation clarifies the categorization of wards used in our study and addresses your concerns regarding the definitions provided in Tables 3 and 4. Thank you once again for your valuable feedback and for the opportunity to enhance the clarity of our manuscript.

Comment 31: In line 176 and 177, please define the final frequency of MRSA among S. aureus in this study. MRSA is confirmed by both disc diffusion and the presence of mecA gene.

Reply: Thank you for your valuable feedback on the final frequency of MRSA among S. aureus isolates in our study. We have revised the manuscript to clearly define the final frequency of MRSA among the S. aureus isolates. Both disc diffusion and the presence of the mecA gene confirmed the final MRSA frequency among the S. aureus isolates in our study, which found out to be 65.1% (56 out of 86 isolates). We have updated lines 176 and 177 in the manuscript to reflect this information accurately.

Comment 32: In table 3 and 4, please justify that why P- value was calculated?

Reply: Thank you for your insightful feedback on our manuscript and for seeking clarification regarding the calculation of p-values in Tables 3 and 4. In our study, p-values were calculated to determine the statistical significance of the observed differences or associations between variables. In Tables 3 and 4, the p-values provide a quantitative measure to evaluate the null hypothesis, which typically states that there is no effect or no difference. The p-values in Tables 3 and 4 wer

---

## [Decision Letter · Decision Letter 1]

22 Jul 2024

Biofilm formation, agr typing and antibiotic resistance pattern in methicillin-resistant Staphylococcus aureus isolated from hospital environments

PONE-D-24-11847R1

Dear Dr. Islam,

We’re pleased to inform you that your manuscript has been judged scientifically suitable for publication and will be formally accepted for publication once it meets all outstanding technical requirements.

Kind regards,

Seyed Mostafa Hosseini

Academic Editor

PLOS ONE

Additional Editor Comments (optional):

Reviewers' comments:

Reviewer's Responses to Questions

**Comments to the Author**

1. If the authors have adequately addressed your comments raised in a previous round of review and you feel that this manuscript is now acceptable for publication, you may indicate that here to bypass the “Comments to the Author” section, enter your conflict of interest statement in the “Confidential to Editor” section, and submit your "Accept" recommendation.

Reviewer #1: All comments have been addressed

Reviewer #2: All comments have been addressed

2. Is the manuscript technically sound, and do the data support the conclusions?

Reviewer #1: (No Response)

Reviewer #2: Yes

3. Has the statistical analysis been performed appropriately and rigorously? 

Reviewer #1: (No Response)

Reviewer #2: Yes

4. Have the authors made all data underlying the findings in their manuscript fully available?

Reviewer #1: (No Response)

Reviewer #2: Yes

5. Is the manuscript presented in an intelligible fashion and written in standard English?

Reviewer #1: (No Response)

Reviewer #2: Yes

6. Review Comments to the Author

Reviewer #1: (No Response)

Reviewer #2: For readers to understand better, it is suggested that the "medicine ward" in Table No. 3 be explained in the manuscript text or below the table.

7. PLOS authors have the option to publish the peer review history of their article (what does this mean?). If published, this will include your full peer review and any attached files.

Reviewer #1: No

Reviewer #2: No

---

## [Editor Report · Acceptance letter]

25 Jul 2024

PONE-D-24-11847R1 

PLOS ONE

Dear Dr. Islam, 

I'm pleased to inform you that your manuscript has been deemed suitable for publication in PLOS ONE. Congratulations! Your manuscript is now being handed over to our production team.

Kind regards, 

on behalf of

Dr. Seyed Mostafa Hosseini 

Academic Editor

PLOS ONE